# Metabolic Alteration Analysis of Steroid Hormones in Niemann–Pick Disease Type C Model Cell Using Liquid Chromatography/Tandem Mass Spectrometry

**DOI:** 10.3390/ijms23084459

**Published:** 2022-04-18

**Authors:** Ai Abe, Masamitsu Maekawa, Toshihiro Sato, Yu Sato, Masaki Kumondai, Hayato Takahashi, Masafumi Kikuchi, Katsumi Higaki, Jiro Ogura, Nariyasu Mano

**Affiliations:** 1Faculty of Pharmaceutical Sciences, Tohoku University, 1-1 Seiryo-machi, Aoba-ku, Sendai 980-8574, Japan; ai.abe.e8@tohoku.ac.jp (A.A.); masafumi.kikuchi.b2@tohoku.ac.jp (M.K.); mano@hosp.tohoku.ac.jp (N.M.); 2Department of Pharmaceutical Sciences, Tohoku University Hospital, 1-1 Seiryo-machi, Aoba-ku, Sendai 980-8574, Japan; toshihiro.sato@tohoku.ac.jp (T.S.); yu.sato.e7@tohoku.ac.jp (Y.S.); masaki.kumondai.d5@tohoku.ac.jp (M.K.); hayato.takahashi.e7@tohoku.ac.jp (H.T.); jiro.ogura@med.id.yamagata-u.ac.jp (J.O.); 3Division of Functional Genomics, Research Centre for Bioscience and Technology, Faculty of Medicine, Tottori University, 86 Nishi-cho, Yonago 683-8503, Japan; kh4060@tottori-u.ac.jp

**Keywords:** Niemann–Pick disease type C, steroid hormones, LC–MS/MS, mitochondria, ammonium solution, metabolic change

## Abstract

Niemann–Pick disease type C (NPC) is an autosomal recessive disease caused by a functional deficiency of cholesterol-transporting proteins in lysosomes, and exhibits various clinical symptoms. Since mitochondrial dysfunction in NPC has recently been reported, cholesterol catabolism to steroid hormones may consequently be impaired. In this study, we developed a comprehensive steroid hormone analysis method using liquid chromatography/tandem mass spectrometry (LC–MS/MS) and applied it to analyze changes in steroid hormone concentrations in NPC model cells. We investigated the analytical conditions for simultaneous LC–MS/MS analysis, which could be readily separated from each other and showed good reproducibility. The NPC phenotype was verified as an NPC model with mitochondrial abnormalities using filipin staining and organelle morphology observations. Steroid hormones in the cell suspension and cell culture medium were also analyzed. Steroid hormone analysis indicated that the levels of six steroid hormones were significantly decreased in the NPC model cell and culture medium compared to those in the wild-type cell and culture medium. These results indicate that some steroid hormones change during NPC pathophysiology and this change is accompanied by mitochondrial abnormalities.

## 1. Introduction

Niemann–Pick disease type C (NPC) is an autosomal recessive disorder characterized by functional deficiencies affecting lysosomal cholesterol transport [1]. These deficits originate from a mutation in the *NPC1* [2,3] or *NPC2* genes [4,5], resulting in lipid accumulation [3,6] and the manifestation of various symptoms [1,7,8,9]. The main symptoms associated with the systemic and central nervous system include hepatosplenomegaly, cholestasis, lung lesions, epilepsy, cognitive dysfunction, and ataxia [10]. Neuropathologically, it is characterized by progressive degeneration of nerve cells, particularly cerebellar Purkinje cells, and foam macrophage infiltration [11]. NPC therapeutic agents, such as miglustat and 2-hydroxypropyl-β-cyclodextrin, have been developed, and although the use of miglustat has been approved in some countries [12], 2-hydroxypropyl-β-cyclodextrin is still undergoing clinical trials [13]. Thus, despite various drug discovery approaches [14,15,16,17], clinical drug development has not yet been achieved. To date, the NPC pathology remains ambiguous, highlighting the necessity to explicate the associated underlying molecular mechanisms in an effort to overcome this disease.

Numerous studies have recently explored mitochondrial abnormalities in NPC. Briefly, these findings indicated that NPC-related lipid accumulation resulted in impaired autophagy, eventuating in abnormal mitochondrial activity [18,19,20,21]. Additionally, increased mitochondrial oxidative stress [22], downregulated ATP production [23], and reduced mitochondrial membrane potential have also been reported [24].

Steroid hormones are vital bioactive metabolites derived from cholesterol, and are synthesized in the endoplasmic reticulum and mitochondria [25,26,27]. These hormones are classified according to their function and structure as glucocorticoids, mineralocorticoids, estrogens, progestins, androgens, and neurosteroids [28,29].

Considering the mitochondrial abnormalities associated with NPC, an abnormal steroid hormone metabolism may be expected. Although some studies have reported an association between several steroid hormones and NPC [30,31,32,33,34,35,36,37], none have comprehensively analyzed the associated changes in steroid hormone metabolism. Therefore, analyzing metabolic changes in NPC may provide further understanding of NPC pathophysiology.

Although radio- and immunoassays are typically used in clinical laboratories to measure steroid hormone concentrations [38], they are limited by the occurrence of cross-reactivity and the inability to perform simultaneous analyses. Liquid chromatography–tandem mass spectrometry (LC–MS/MS) is a highly sensitive method utilized for comprehensive analyses [39] and would provide additional analytical benefits since the amount of steroid hormones in cells or body fluids are very small. Furthermore, although many LC–MS/MS techniques combined with derivatization have been reported [40,41], they are often unsuitable for the comprehensive analysis of steroid hormones, which possess a variety of functional groups.

In this study, we aimed to analyze the metabolic alterations of steroid hormones in NPC and develop a simultaneous steroid hormone analysis method using LC–MS/MS, thereby allowing for a deeper understanding of the NPC pathophysiology and its involvement in mitochondrial steroid hormone production. A gonad-derived NPC cell model was used in this study [42]. First, simultaneous analysis conditions were developed for 13 steroid hormones (Figure 1). ESI–MS/MS, LC, and pretreatment conditions were investigated and the analytical method was validated. Next, we verified the characteristics and morphology of NPC model cells. Finally, we analyzed the NPC model and wild-type cells and compared their steroid hormone levels.

## 2. Results and Discussion

### 2.1. MS/MS Optimization for Steroid Hormone Analysis

LC–MS/MS is capable of simultaneous steroid hormone analyses, thus providing a suitable platform for the development of a comprehensive protocol targeted for these hormone concentrations [39,40,41,43,44,45,46,47,48,49]. Although derivatization by conjugating proton affinity groups to analytes is widely utilized [40,41,43,44,45,46,47,48], LC–MS/MS analysis methods without derivatization have recently been reported [49,50,51]. Moreover, Li et al. and Gaudl et al. utilized ammonium fluoride as an additive, thereby allowing for the simultaneous analysis of various estrogens [49,51]. Considering this, the current study was aimed at developing a comprehensive steroid hormone-specific analytical method using LC–MS/MS without derivatization. MS parameters that were optimized via the infusion of standard solutions included declustering potential (DP), collision energy (CE), and cell exit potential (CXP). The MS polarity was separated into positive and negative ion detection modes in a similar manner as described in previous reports [50,51]. A total of ten and three compounds were observed in positive and negative ion detection modes, respectively (Table 1). These base peaks subsequently served as precursor ions, and the most intense selected reaction monitoring (SRM) transitions and CE values were optimized accordingly and are presented in Table 1.

Mobile phase additives, such as ammonium solutions [52,53,54], were investigated in the negative ion detection mode to increase analysis sensitivity. The aqueous ammonium solution exhibited the most intense peaks (Figure 2). Considering that the pH of the ammonium aqueous solution [52,53,54] and ammonium fluoride solution [49,51] is approximately 11 and the pKa of the phenolic groups on estrogens is approximately 10.8 [55], these solutions negatively ionize more phenolic groups, thereby producing stronger ion intensities than neutral or acidic pH solutions.

### 2.2. LC optimization for Steroid Hormone Analysis

Since the structural isomers in this study exhibited similar SRM transitions, we investigated the LC conditions for isomer separation. Reversed-phase chromatography is the most suitable method for LC separation of steroid hormones. Various stationary phase columns, such as phenyl-hexyl [50], poly[styrenevinylbenzene] particle (POROSTM column) [51], and octadecylsilyl (ODS) columns have previously been used [49,56]. The ODS-bonded column was utilized in the current study, considering that it allows good retention of steroid hormones. Furthermore, two simultaneous LC–MS/MS analysis conditions were examined and developed in the current study; one for the positive and the other for the negative ion detection modes.

An ODS-bonded core-shell silica gel was used as the analytical column for LC separation and subsequent detection of analytes in the positive ion detection mode, and formic acid was used as an additive under acidic conditions. Several gradient conditions were investigated and revealed that good separation was achieved when the mobile phase B concentration increased from 35% to 80% (Figure 3a). The associated retention times are listed in Table 1.

Next, the LC conditions were optimized for the negative ion detection. A highly sensitive analysis was performed at pH 11 using an aqueous ammonium solution as an additive for MS–MS optimization. However, considering that silica particles of silanol group are disrupted under basic conditions, general ODS-bonded silica columns cannot be used. Therefore, we utilized a Triart C18 Plus, an ethylene crosslinked-hybrid silica gel column in which the basicity and pressure resistance of the silica gel were improved. The LC gradient separation conditions required for separating estrone and estradiol, while retaining the estriol, were evaluated. Good LC separation was achieved when the mobile phase B concentration increased from 40% to 75% in 7 min (Figure 3b).

### 2.3. Extract Recovery

Pretreatment, such as solid-phase extraction [49,50,51] and liquid–liquid extraction [45,47,48,56], is imperative when analyzing endogenous compounds in biological samples. Although solid-phase extraction is a simple sample pretreatment method, the required solid-phase cartridges are expensive. Contrarily, liquid–liquid extraction allows for effortless removal of contaminants using two immiscible liquids. Considering that all analytes and ISs are non-ionized steroids, the latter extraction method can be easily applied in the context of this study. As such, ethyl acetate (AcOEt) and tert-butyl methyl ether (tBME) were used as the organic layer while water was used as the aqueous layer. Furthermore, five combinations of organic solvents were investigated and the absolute recovery rate was used for performance evaluation. The latter evaluation revealed that highly stable recovery of steroid hormones was achieved on performing liquid–liquid extraction twice with AcOEt (Appendix A).

### 2.4. Analytical Method Validation

To effectively quantify metabolites in a biological sample, it is important to ensure good reliability of the analytical method. Therefore, validation of the developed method is required to confirm the validity and reliability of the process. In this section, we verify the matrix effect, calibration curve linearity, and reproducibility to evaluate the reliability of the analysis method by using cells and media. In the analysis of endogenous substances, it is difficult to obtain a blank matrix without utilizing steroids. As such, the validity of this analytical method was evaluated based on previous reports [57,58,59,60,61].

#### 2.4.1. Matrix Factor

The matrix effect of the analytical method, which was evaluated for each matrix of the cells and medium, is a combination of the recovery rate and extraction process efficiency. Matrix factor (MF) [57,58,62] and the IS-normalized MF were calculated to evaluate the matrix effect [57,58,62]. Several MFs were approximately 100 ± 15% in both the cells and medium (Appendix A). Although some analytes showed >15% matrix effects (epiandrosterone on medium and estrone, estradiol, estriol, estrone-^2^H_4_, estradiol-^13^C_3_, and estriol-^13^C_3_ in cells), their MFs were normalized within 100 ± 15% using their ISs (Appendix A). In other words, the matrix effect on the quantification of all analytes was compensated for by their ISs.

#### 2.4.2. Calibration Curves

Calibration curves were constructed for the quantification of steroid hormones. Considering that the analytical methods were unaffected by the matrix, as discussed in Section 2.4.1, water was used as a surrogate matrix that did not include the analytes. Calibration curves were prepared using each IS, all of which showed good linearity (Appendix A).

#### 2.4.3. Reproducibility Test

Quality control (QC) samples, which were prepared by spiking the matrix with standard solutions, were quantified using calibration curves. The accuracy and precision of the method was evaluated by using the relative error (RE) and the coefficient of variation (CV), respectively. Considering the low analyte concentrations present in the matrix, the calculations were performed to accommodate their endogenous amounts. Thus, accuracy was evaluated based on the spiked QC samples. In the cell samples, aldosterone, androsterone, and epiandrosterone levels were >±15% (Appendix A) and estrogen levels were within ±15% in all assays (Appendix A). In the medium samples, aldosterone, testosterone, androsterone, and epiandrosterone levels were >±15% (Appendix A) and estrogen levels were within ±15% in all assays (Appendix A). An important consideration is that, for analytes presenting with larger errors, LC–MS/MS-calculated steroid hormone concentrations may differ from their actual values.

### 2.5. Cells

Because steroid hormones exhibit sex-related differences in their expression and diurnal changes, NPC model cells were used to analyze the metabolic changes associated with steroid metabolism in NPC pathology. Two causative genes, namely *NPC1* and *NPC2*, are mutated in patients with NPC [1,7,9,63]. Considering that 95% of NPC cases present with mutation in the *NPC1* gene [1,7,9,64], the *NPC1* mutant cell is often used as an NPC model cell [42,64]. In this study, we used an *Npc1*-deficient Chinese hamster ovary (CHO) cell line established using gene trap mutagenesis as an NPC model cell [42]. The characteristics of the NPC model cells were verified before LC–MS/MS analysis.

#### 2.5.1. Confirmation of Cholesterol Accumulation Using Filipin Staining

Since NPC patients lack functional cholesterol-transporting proteins [1,6], lysosomal cholesterol accumulates in cells [65,66,67]. Therefore, we confirmed unesterified cholesterol accumulation in NPC model cells (Appendix A) using filipin staining [65,66,67,68], which is a standard and conclusive diagnostic method for NPC [7,9]. As such, it was confirmed that the model cells used here successfully represented NPC.

#### 2.5.2. Organelle Observation

Organelle morphology of the NPC model cells was confirmed via staining with Organelle-ID-RGB, a mixed reagent containing three fluorescent compounds for mitochondria, nuclei, and lysosomes. The fluorescence signal for mitochondrial morphology (green) indicated that the wild-type cells were rod-shaped (Appendix A). In agreement with previous reports [18,20], numerous spherical mitochondria were observed in the NPC model cells and fragmentation was also confirmed (Appendix A). In addition, lysosome (red) and mitochondria (green) localization were observed to be complementary (Appendix A). In other words, the mitochondria were involved in the lysosomal degradation process and presented incomplete mitochondrial morphology associated with NPC, which is consistent with previous reports [20,69,70].

### 2.6. Steroid Hormone Analysis in NPC Model Cell

Finally, steroid hormones in the NPC cells and the medium were analyzed, and the associated metabolic changes were investigated. The cell culture and sample collection scheme is summarized in Appendix A. Studies on the in vitro metabolism of steroid hormones have often used media as samples [71,72]. Considering that phenol red has estrogen-like action [73,74], the current study utilized medium that excluded phenol red. In addition, fetal bovine serum (FBS) acts as a nutrient and contains steroid hormones. As such, and based on previous reports [75,76], the 10% FBS-supplemented medium was replaced with FBS-free medium in the current study. Furthermore, both medium [77,78,79,80,81,82,83] and cells [84] have previously been used as samples. Following analysis, it was observed that testosterone, androsterone, progesterone, and estrone levels were significantly decreased in the NPC model cells (Figure 4 and Appendix A). Moreover, testosterone levels were particularly increased in the NPC model compared to the wild-type cells, while androsterone levels were significantly decreased in the NPC cells.

In contrast, epiandrosterone, dehydroepiandrosterone (DHEA), glucocorticoid, and aldosterone levels were not significantly different between two cells. Furthermore, progesterone levels significantly decreased in NPC cells (wild-type: 4.27 ± 2.87 pg/10^6^ cells, NPC model cells: 0.342 ± 0.148 pg/10^6^ cells), while pregnenolone levels did not change. Among the estrogens, estrone levels significantly decreased in NPC cells (wild-type: 0.166 ± 0.00918 pg/106 cells, NPC model cells: 0.0483 ± 0.0120 pg/106 cells), estradiol levels were not significantly different, and estriol was only detectable in the NPC model cells, in comparison to their levels in the wild-type cells.

Medium analysis indicated the presence of steroid hormones, even in FBS-free medium without cell cultures. Therefore, we analyzed both fresh and 48 h cell culture media and subsequently calculated the difference in steroid hormone levels between the two media. A sub-zero (negative) difference in epiandrosterone and corticosterone levels (Figure 5 and Appendix A) indicated that the steroid hormone levels were decreased in the 48 h cell culture medium compared to those in fresh medium. This may have resulted from the chemical conversion of steroid hormones into the medium or, alternatively, their cellular uptake. Steroid hormones are generally chemically stable [85,86], and FBS-free medium contains no metabolic enzymes, suggesting that cellular uptake was the cause. Steroid hormones are passively taken up by cells, not via transporters. Considering that intracellular epiandrosterone and corticosterone concentrations were higher than those of the other hormones (Figure 4 and Appendix A), their uptake may be greater than that of the other steroid hormones. Among the corticosteroids, cortisone levels decreased significantly in NPC model cell, whereas those of the others did not change. Furthermore, progesterone levels decreased significantly among progestins. Among the estrogens, estrone and estradiol levels were significantly decreased. Lastly, estriol, androsterone, and pregnenolone were not detected in the culture media.

In summary, the results of steroid hormone analyses in cells and media indicated that testosterone, androsterone (classified as androgens), progesterone (progestins), as well as estrone and estradiol (classified as estrogens) decreased significantly (Figure 4 and Figure 5, Appendix A). Steroid hormones can be synthesized from both mevalonic acid and exogenous cholesterol [87], explaining their detection even in patients with abetalipoproteinemia [88]. Steroid hormones are synthesized in specific organs, including the adrenal cortex, gonads, and brain [26,36,89]. First, unesterified cholesterol is absorbed into the mitochondria by steroid-producing acute regulatory protein (StAR) [90]. CYPscc subsequently produces pregnenolone [91], which is sequentially metabolized via two mechanisms. The first mechanism involves 3β-hydroxy group oxidation, catalyzed by 3β-hydroxysteroid dehydrogenase (3β-HSD) [92], to produce progesterone. Additionally, 3β-HSD 2 is expressed in ovaries. The second mechanism involves side-chain cleavage with 17α-hydroxylase/17,20 lyase (CYP 17) [93], which produces DHEA [94]. Interestingly, CYP 17 is also expressed in the ovaries [95]. Progesterone is metabolized to deoxycorticosterone [96], which is further metabolized, along with 11-deoxycortisol, by 11β-hydroxylase (CYP11B1) in mitochondria [26]. Corticosterone is then metabolized to aldosterone by aldosterone synthase (CYP11B2) [26]. Moreover, 11-deoxycortisol is metabolized to cortisol and cortisone while aldosterone and cortisol are also produced in the adrenal cortex. In this study, ovarian cells were used. These cells do not generate aldosterone, cortisol, or cortisone; however, these hormones were detected in these cells (Figure 4 and Figure 5, Appendix A). It is possible that the duration of cell culture in FBS-free medium might have been insufficient. DHEA and androstenedione are metabolized to androstenediol and testosterone, respectively, by 17β-HSD [97], and 17β-HSD 2 is expressed in the ovaries [98]. Androstendiol is metabolized to testosterone by 3β-HSD, and androstenedione and testosterone are metabolized to estrone and estradiol, respectively, by aromatase (CYP19A). Estriol is produced from 16-hydroxyestrone by both P450 and 17β-HSD [99]. Aromatase, 3β-HSD, and 17β-HSD were expressed in cells used in this study.

We found decreased levels of various steroid hormones (Figure 4 and Figure 5, Appendix A), suggesting that steroid hormone production may be lower in NPC cells than in normal cells. Most steroidogenic enzymes are in the mitochondria or endoplasmic reticulum and some use coenzymes [27]. NAD, for example, is the coenzyme of 3β-HSD, and is generally used for mitochondrial energy production [92,100]. Although both mitochondrial membrane potential and ATP synthesis must be fully functional for the synthesis of steroid hormones [101,102], mitochondrial dysfunction due to NPC pathology was observed in this study (Appendix A). Similarly, patients with early diabetes present with reduced levels of some neurosteroids and neurosteroid-producing enzymes in the brain in addition to mitochondrial dysfunction [103,104,105]. Glucocorticoid levels decrease in nicotinamide nucleotide transhydrogenase (NNT) deficiency [106]. NNT uses transmembrane potential to regenerate NADPH and the mitochondrial antioxidant system to regulate energy metabolism. Therefore, NNT deficiency impairs steroid hormone synthesis by reducing the energy supply [106]. As such, it may be concluded that steroid hormone production is impaired in patients with mitochondrial disorders (Figure 6).

In summary, NPC may involve multiple mitochondrial disorders and the reduction of various steroid hormones. Generally, steroid hormones are secreted into the blood and exert their effects via various nuclear receptors [28,107,108,109]. Estrogens, progesterone, and androgens, also known as sex hormones, are involved in the maintenance of cognitive and reproductive function. Furthermore, neurosteroids, which are synthesized in the brain, affect neuronal growth, survival, and differentiation, [110,111,112] and have neuroprotective effects [110], while some neurosteroids act as neurotransmitters [113]. Purkinje cells are the major neurosteroid production sites, and mammals actively synthesize neurosteroids from cholesterol [114]. However, cerebellar Purkinje neurons die in NPC patients [115]. Although the brain and neurosteroidogenic functions in NPC are unknown, they should be analyzed in the future.

Patients with NPC present with various symptoms, including inflammation and neurological symptoms [1,7,9,116]. Additionally, neurosteroid production may be affected by Purkinje cell degeneration [115]. In the future, we aim to analyze steroid hormones in humans and clarify their effects on NPC pathology.

## 3. Materials and Methods

### 3.1. Chemicals and Reagents

Cortisol and testosterone-[^2^H_3_] were purchased from Sigma-Aldrich (St. Louis, MO, USA). Cortisone, aldosterone, corticosterone, and progesterone were purchased from Steraloids (Newport, RI, USA) and pregnenolone was purchased from Funakoshi (Tokyo, Japan). Dehydroepiandrosterone, epiandrosterone, and androsterone were obtained from Tokyo Chemical Industry Co., Ltd., (Tokyo, Japan). Estrone, estradiol, estriol, acetic acid (special grade reagent), tBME, and filipin complex were purchased from Nacalai Tesque (Kyoto, Japan). Progesterone-[^2^H_9_], 17β-estradiol-[^13^C_3_], aldosterone-[^2^H_7_], and estrone-[^2^H_4_] were acquired from ISOTEC (Miamisburg, OH, USA). Cortisol-[^13^C_3_], androsterone-[^2^H_4_], estriol-[^13^C_3_], dehydroepiandrosterone-[^2^H_6_], and pregnenolone-[^13^C_2_,^2^H_2_] were purchased from Cambridge Isotope Laboratories (Tewksbury, MA, USA). Testosterone, formic acid (special grade reagent), 28% aqueous ammonia solution (special grade), and the wako free cholesterol E-test were obtained from Fuji Film Wako Pure Chemical Industries (Osaka, Japan). First-grade methanol and AcOEt were first distilled following their purchase from Fuji Film Wako Pure Chemical Industries (Osaka, Japan). HPLC-grade acetonitrile was obtained from Kanto Chemical Co., Inc. (Tokyo, Japan). Ultrapure water was purified using a Puric-α purification system (ORGANO CORPORATION, Tokyo, Japan). In addition, Organelle-ID-RGB^®^ Reagent I was acquired from Enzo Life Sciences Inc. (Farmingdale, NY, USA).

### 3.2. LC–MS/MS Equipment

A QTRAP 6500 linear ion trap-quadrupole hybrid tandem mass spectrometer (SCIEX, Framingham, MA, USA) equipped with an ESI probe was connected to a Nexera series ultra-high-performance liquid chromatograph (Shimadzu, Kyoto, Japan). Measurements were performed in positive or negative ion detection modes, while quantitative analysis was performed via measurement and peak processing using Analyst 1.6.2 and SCIEX OS-Q (SCIEX).

### 3.3. ESI–MS/MS Conditions

In both the positive and negative ion detection modes, 1 μg/mL of each standard solution was infused at a flow rate of 10 μL/min, while the mobile phases A and B were delivered at a flow rate of 390 μL/min and a ratio of 2:3. Optimization of each SRM condition was established by analyte diluting with the mobile phases, and the optimizations are listed in Table 1. Additionally, the optimized ion source parameters used in each mode are listed in Appendix A.

### 3.4. LC Conditions

#### 3.4.1. Positive Ion Detection Mode

Gradually changing ratios of formic acid/water (0.1:100, *v*/*v*) and formic acid/methanol/acetonitrile (0.1:50:50, *v*/*v*/*v*) served as mobile phases A and B, respectively (Appendix A). A flow rate of 0.4 mL/min was applied in a CAPCELL CORE C18 column (2.1 mm inner diameter × 150 mm, 2.7 μm; Osaka Soda; Osaka, Japan) set at 40 °C. A sample volume of 50 µL was injected into the LC–MS/MS system for analysis.

#### 3.4.2. Negative Ion Detection Mode

Mobile phases A and B were comprised of 28% aqueous ammonia/water (0.1:100, *v*/*v*) and 28% aqueous ammonia/methanol/acetonitrile (0.1:50:50, *v*/*v*/*v*), respectively, and were delivered in gradient mode at a flow rate of 0.4 mL/min. Triart C18 Plus (2.1 mm inner diameter × 150 mm, 3 μm; YMC; Kyoto, Japan) was utilized at a temperature of 40 °C. A sample volume of 50 μL was injected, and the obtained gradient conditions are listed in Appendix A.

### 3.5. Preparation of Standard Steroid Hormone Solutions

The standard reagents were precisely weighed for each steroid hormone and subsequently dissolved in ethanol, obtaining a concentration of approximately 100 μg/mL. Each standard stock solution was mixed and diluted with ethanol and subsequently utilized for calibration curve construction and QC purposes. The standard solution for the calibration curve was prepared in the range of 1 pg/mL to 100 ng/mL (detailed in Section 3.9). The standard solutions utilized for QC purposes during positive ion detection included QCL (0.6 ng/mL), QCM (6 ng/mL), and QCH (50 ng/mL), while those for negative ion detection included QCL (0.06 ng/mL), QCM (0.6 ng/mL), and QCH (50 ng/mL). All standards were stored at −20 °C.

### 3.6. Preparation of IS Solution

Compounds labeled with stable isotopes were precisely weighed and dissolved in ethanol to obtain a concentration of 10 µg/mL. Each standard solution was further diluted with ethanol to prepare a 2 ng/mL mixed solution. Subsequently, each standard stock solution was diluted with AcOEt to prepare a mixed IS solution. The 17β-estradiol-[^13^C_3_], aldosterone-[^2^H_7_], estrone-[^2^H_4_], progesterone-[^2^H_9_], cortisol-[^13^C_3_], and estriol-[^13^C_3_] solutions were diluted to 30 pg/mL, whereas the androsterone-[^2^H_4_], dehydroepiandrosterone-[^2^H_6_], and pregnenolone-[^13^C_2_,^2^H_2_] solutions were diluted to 300 pg/mL. All prepared standards were stored at −20 °C.

### 3.7. Liquid–Liquid Extraction

Following addition of 1 mL IS solution diluted in AcOEt and 3 mL AcOEt to 2 mL of sample solution (medium or cell suspension), the mixtures were centrifuged at 3000× *g* for 10 min, and the separated upper layer was transferred to another tube. Centrifugation was repeated following addition of 4 mL AcOEt to the lower layer, and the resulting upper layer was extracted again. The upper layers obtained using the two liquid–liquid extractions were combined, evaporated under a nitrogen stream, and re-dissolved in 100 µL water/methanol (50:50, *v*/*v*). Of these prepared samples, 50 µL aliquots were injected for analysis in both the positive and negative ion detection modes.

### 3.8. Matrix Effect

To evaluate the matrix effect, MF and IS-normalized MF was determined following the introduction of standards to the matrix using the calculations depicted below [57,60,61,117]. Briefly, 2 mL of cell suspension containing 8.6 × 10^6^ cells or medium was used as the matrix, and 100 μL of 2 ng/mL standard solution and 100 μL of 2 ng/mL IS solution was added to prepare three sample sets (A, B, and C). Sample set A comprised standard and IS mixed solutions without a cellular matrix. Sample set B comprised the standard solution and the cellular matrix. Sample set C comprised only the cellular matrix. All sample sets were analyzed, and the peak areas of the analytes and the IS were integrated. The absolute recovery, MF (%), is the combined result of the system recovery rate and matrix effect. The calculated IS-normalized MF is the ratio of the MF to the IS, and was used to evaluate the effect of the measurement method matrix.
$$\mathrm{Matrix}\;\mathrm{factor}\;\left(\%\right)=\frac{\left(\mathrm{peak}\;\mathrm{area}\;\mathrm{of}\;\mathrm{sample}\;\mathrm{set}\;B-\mathrm{peak}\;\mathrm{area}\;\mathrm{of}\;\mathrm{sample}\;\mathrm{set}\;C\right)}{\left(\mathrm{peak}\;\mathrm{area}\;\mathrm{of}\;\mathrm{sample}\;\mathrm{set}\;A\right)}\times 100$$
$$\mathrm{IS}\;\mathrm{normalized}\;\mathrm{matrix}\;\mathrm{factor}\;\left(\%\right)=\frac{\left(\mathrm{matrix}\;\mathrm{factor}\;\mathrm{of}\;\mathrm{analytes}\right)}{\left(\mathrm{matrix}\;\mathrm{factor}\;\mathrm{of}\;\mathrm{IS}\right)}\times 100$$

### 3.9. Calibration Curve Preparation

The standard solution was mixed and diluted to 1, 3, 6, 10, 30, 60, 100, 300, and 600 pg/mL, and 1, 3, 6, 10, 30, 60, and 100 ng/mL. After drying 100 μL of the standard solution under a light stream of nitrogen, water (0.5 mL) was added as an alternative matrix, and 1 mL IS solution in AcOEt was added and re-dissolved. The solution was centrifuged at 14,000 rpm for 5 min and the separated upper layer was recovered. Thereafter, 1 mL AcOEt was added to the lower layer, and the centrifugation step was repeated. The upper layers obtained were combined and dried under a nitrogen stream. The standards were re-dissolved in 100 µL water/methanol (1:1, *v*/*v*) and subjected to both positive and negative mode LC–MS/MS analysis (50 µL per analysis). The peak area ratio of the measurement target and IS was plotted at each concentration and a calibration curve was constructed using the least-squares method with 1/x^2^ weighting.

### 3.10. Reproducibility Test

QC standard solutions of steroid hormones were dried under a nitrogen stream. Subsequent to the addition of 0.5 mL medium or cells in PBS, 1 mL AcOEt, and 1 mL IS solution in AcOEt, the solution was centrifuged at 14,000 rpm for 5 min to extract the upper layer. Thereafter, 2 mL of AcOEt was added to the lower layer, the centrifugation step was repeated, and the upper layer was re-extracted. The two obtained upper layers were combined, dried under a nitrogen stream, and re-dissolved in 100 μL of 50% methanol. Lastly, 50 µL of each prepared solution was injected for reproducibility measurements.

Three QC standard solutions were prepared for positive (10 compounds) and negative (3 compounds) ion detection mode as LQC (0.3 ng/mL and 0.03 ng/mL, respectively), MQC (3 ng/mL and 0.3 ng/mL, respectively), and HQC (80 ng/mL for both modes).

These QC samples were analyzed using a calibration curve in the same analysis batch (each concentration, *n* = 3). The QC concentration and formula used were as follows:$$\mathrm{RE}\;\left(\%\right)=\frac{(\mathrm{mean}\;\mathrm{value}\;\mathrm{of}\;\mathrm{each}\;\mathrm{QC}\;\mathrm{value}-\left[\mathrm{QC}\;\mathrm{concentration}+\mathrm{endougenous}\;\mathrm{concentration}\right)}{\left(\mathrm{QC}\;\mathrm{concentration}+\mathrm{endogenous}\;\mathrm{concentration}\right)}$$
$$\mathrm{CV}\;\left(\%\right)=\frac{\mathrm{standard}\;\mathrm{deviation}\;\mathrm{of}\;\mathrm{quantitative}\;\mathrm{value}}{\mathrm{mean}\;\mathrm{value}\;\mathrm{of}\;\mathrm{quantitative}\;\mathrm{value}}\times 100$$

### 3.11. NPC Model Cell and Medium

We used the *Npc1* gene to trap CHO cells as NPC model cells [44], while wild-type CHO cells were used as controls. Dulbecco’s modified Eagle’s medium (DMEM)/Ham F-12 medium (L-glutamine, pyruvic acid, HEPES-free, phenol red-free) was purchased from Nacalai Tesque and subcultured with deactivated 10% FBS.

### 3.12. Cholesterol Staining

The filipin staining solution was prepared by mixing filipin (0.5 mg), dimethyl sulfoxide (25 μL), and 10% FBS-supplemented PBS. The cells (2.0 × 10^5^ cells/well) were seeded in 12-well plates on a cover glass. After reaching 90% confluence, the medium was removed, and the plates were washed with ice-cold PBS before being incubated with 200 μL of 4% paraformaldehyde at 25 °C for 20 min. After washing three times with ice-cold PBS, the plates were incubated with 600 μL of filipin staining solution at 25 °C for 30 min. The plates were washed again twice with ice-cold PBS, and 250 μL PBS was added. The cover glass was then removed, placed on a glass slide, and fixed using an adhesive. Slides were placed under a BZ9000 fluorescence microscope (Keyence, Osaka, Japan), and the cellular unesterified cholesterol was observed using DAPI-BP as a fluorescence filter.

### 3.13. Organelle Observation

Cover glasses were laid on 12-well plates for cell culture and washed twice with 1 mL ethanol and medium. Thereafter, 2.0 × 10^5^ cells were seeded per well and grown to 90% confluence. The nuclei, lysosomes, and mitochondria were simultaneously stained using Organelle-ID-RGB^®^ reagent I. The staining solution was prepared by diluting 2 µL of the staining reagent with 1 mL PBS. Before staining, the medium was removed, 250 μL of staining solution was added, and the cells were incubated at 37 °C for 30 min. After removal of the staining solution, the cells were washed twice with 250 μL of PBS, and 250 μL of PBS was added. The cover glass was removed, placed on a glass slide, and fixed using an adhesive. Cell and organelle morphology were observed using a BZ9000 fluorescence microscope under a 100-fold oil-immersed objective lens. DAPI-BP (excitation [Ex], 360 nm; emission [Em], 460 nm), GFP-BP (excitation, 470 nm; emission, 535 nm), and TRITIC (excitation, 540 nm; emission, 605 nm) were used as fluorescence filters. The shapes of nuclei, mitochondria, and lysosomes were also observed.

### 3.14. Cells for LC–MS/MS Analysis of Steroid Hormones

The cells (2.0 × 10^6^) were seeded in a 100 mm petri dish and grown in a medium containing 10% FBS. After 24 h, the medium was replaced with FBS-free medium, and the cells were collected after another 48 h. After collecting the medium, the cells were washed twice with 2 mL ice-cold PBS and were subsequently collected from the petri dish using a scraper. Lastly, the suspended cells were counted and analyzed using LC–MS/MS.

### 3.15. Sample Preparation Procedure

To 2 mL cell suspension containing either 8.6 × 10^6^ cells or 2 mL cell-culturing medium, 1 mL AcOEt and 3 mL IS solution in AcOEt were added, and the suspension was then centrifuged at 3000 rpm for 10 min to separate the upper layer. After fractionation of the upper layer, 4 mL of AcOEt was added to the lower layer, and liquid–liquid extraction was repeated. The two upper layers were combined, dried under nitrogen gas, and redissolved in 100 μL of 50% methanol. Each 50 μL aliquot was subjected to LC–MS/MS analysis for each polarity.

### 3.16. Quantification of Steroid Hormones in the Sample

Steroid hormones in the cells and medium were analyzed using LC–MS/MS. The peak area of the SRM chromatogram was integrated and quantified using the peak area ratio of each compound relative to the IS.

The amount of steroid hormone in the cell suspension was calculated using the following formula:$$\begin{array}{l} \mathrm{Amount}\;\mathrm{in}\;\mathrm{cell}\;\left(\mathrm{pg}/10^{6}\;\mathrm{cells}\right)\\ \;\;\;\;\;\;\;\;=\left(\mathrm{quantified}\;\mathrm{steroid}\;\mathrm{hormones}\;\mathrm{in}\;\mathrm{cell}\;\mathrm{suspension}\;\left[\mathrm{pg}/\mathrm{mL}\right]\times \frac{\mathrm{cell}\;\mathrm{suspension}\;\mathrm{volume}\;\left[\mathrm{mL}\right]}{\mathrm{cell}\;\mathrm{count}\;\left[10^{6}\;\mathrm{cells}\right]}\right) \end{array}$$

The difference between steroid hormones in the cell cultures were calculated using the following equation:(1)$$\begin{array}{l} \mathrm{The}\;\mathrm{difference}\;\mathrm{in}\;\mathrm{medium}\;\left(\mathrm{pg}/10^{6}\;\mathrm{cells}\right)\\ \;\;\;\;\;\;\;=\{(\mathrm{quantified}\;\mathrm{steroid}\;\mathrm{hormone}\;\mathrm{in}\;\mathrm{cell}\;\mathrm{culture}\;\mathrm{medium}\;\left[\mathrm{pg}/\mathrm{mL}\right]\\ \;\;\;\;\;\;\;-\mathrm{quantified}\;\mathrm{steroid}\;\mathrm{hormone}\;\mathrm{in}\;\mathrm{fresh}\;\mathrm{medium}\;\left[\mathrm{pg}/\mathrm{mL}\right])\times \frac{\mathrm{medium}\;\mathrm{volume}\;\left(\mathrm{mL}\right)}{\mathrm{cell}\;\mathrm{count}\;\left(10^{6}\;\mathrm{cells}\right)}\} \end{array}$$

### 3.17. Statistical Analysis

All experimental results are presented as mean ± standard deviation (S.D.). To determine statistically significant differences, a Wilcoxon’s test was applied. Differences with * *p* < 0.05 and ** *p* < 0.005 were considered statistically significant.

## 4. Conclusions

In this study, we developed a method for the simultaneous analysis of steroid hormones using LC–MS/MS to evaluate the associated metabolic changes in NPC model cells, thereby allowing for partial deciphering of the NPC pathophysiology. Following evaluation of LC–MS/MS conditions, pretreatment, and analytical method validation, we succeeded in the simultaneous LC–MS/MS analysis of 13 steroid hormones in cells and media. The levels of steroid hormones, such as androgens and estrogens, were decreased in NPC model cells, possibly due to various impaired mitochondrial functions, suggesting an altered steroid hormone metabolism in NPC patients. As such, the current study has illuminated novel aspects of the NPC pathophysiology, which may be used for the discovery of novel NPC treatment targets in future studies.

## Figures and Tables

**Figure 1 ijms-23-04459-f001:**
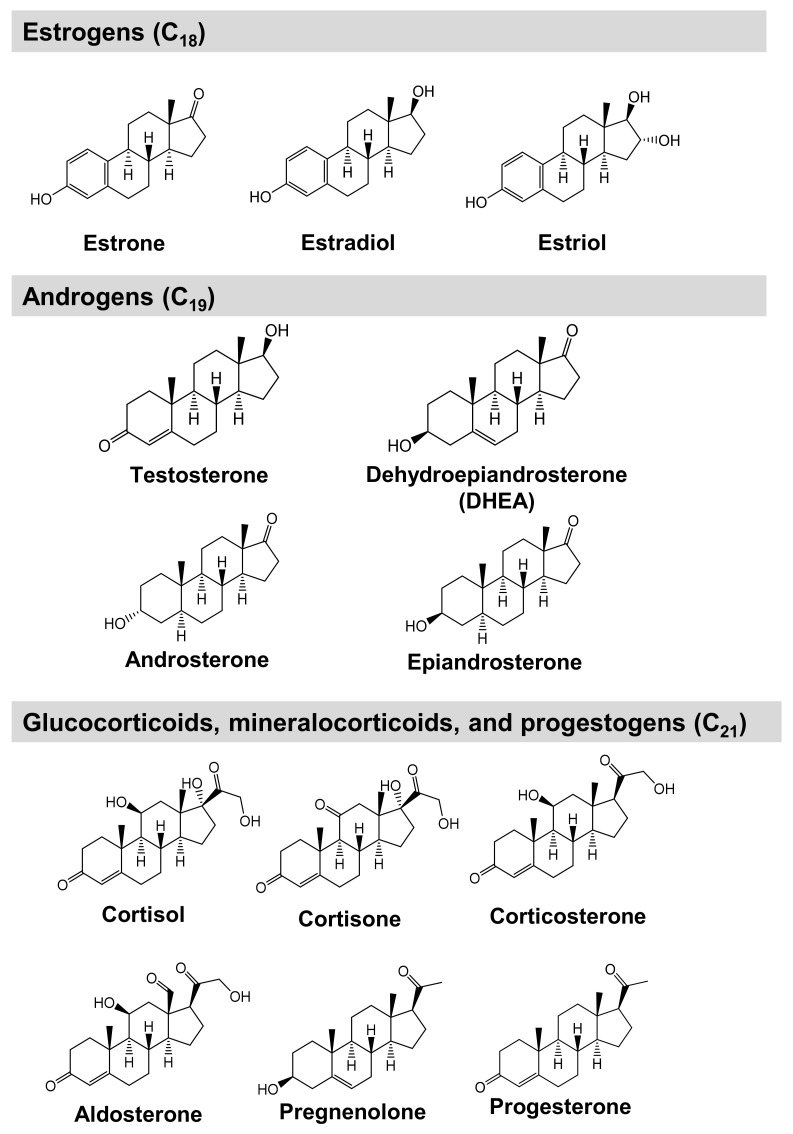
Chemical structures of steroid hormones analyzed in this study.

**Figure 2 ijms-23-04459-f002:**
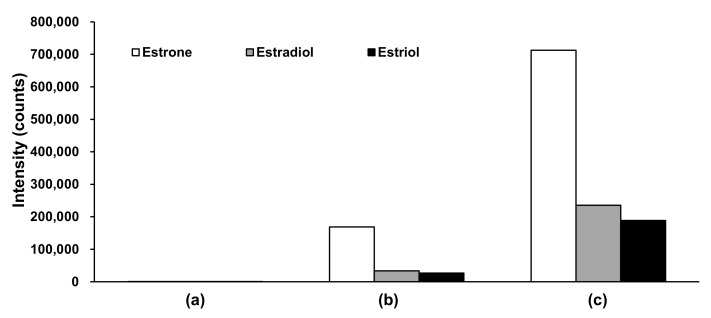
Comparison of peak intensities in negative ion detection mode for various mobile phases. (**a**) Acidic conditions using mobile phase A: Formic acid/water (0.1:100, *v*/*v*) and mobile phase B: Formic acid/MeOH/ACN (0.1:50:50, *v*/*v*/*v*); (**b**) weak basic conditions using mobile phase A: Formic acid/water (0.1:100, *v*/*v*) and mobile phase B: 1 M ammonium acetate aqueous solution/MeOH/ACN (0.1:50:50, *v*/*v*/*v*); (**c**) basic conditions using mobile phase A: 28% ammonium aqueous solution/water (0.1:100, *v*/*v*) and mobile phase B: 28% aqueous ammonium solution/MeOH/ACN (0.1:50:50, *v*/*v*/*v*). ACN, acetonitrile; MeOH, methanol.

**Figure 3 ijms-23-04459-f003:**
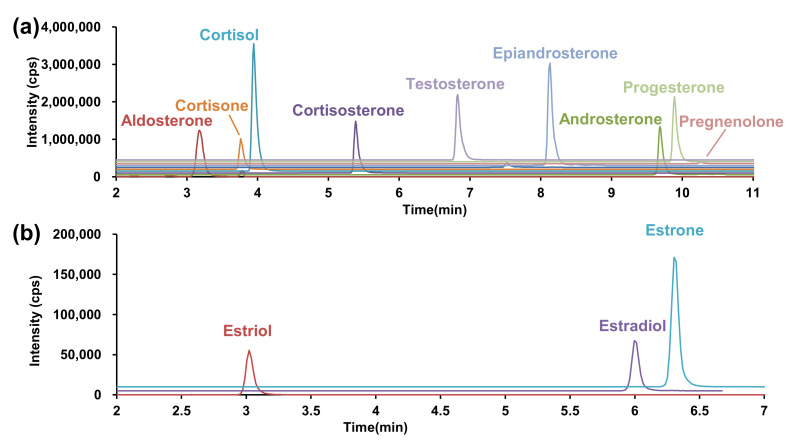
SRM chromatogram under optimized conditions. (**a**) Positive ion detection mode, (**b**) negative ion detection mode. SRM, selected reaction monitoring.

**Figure 4 ijms-23-04459-f004:**
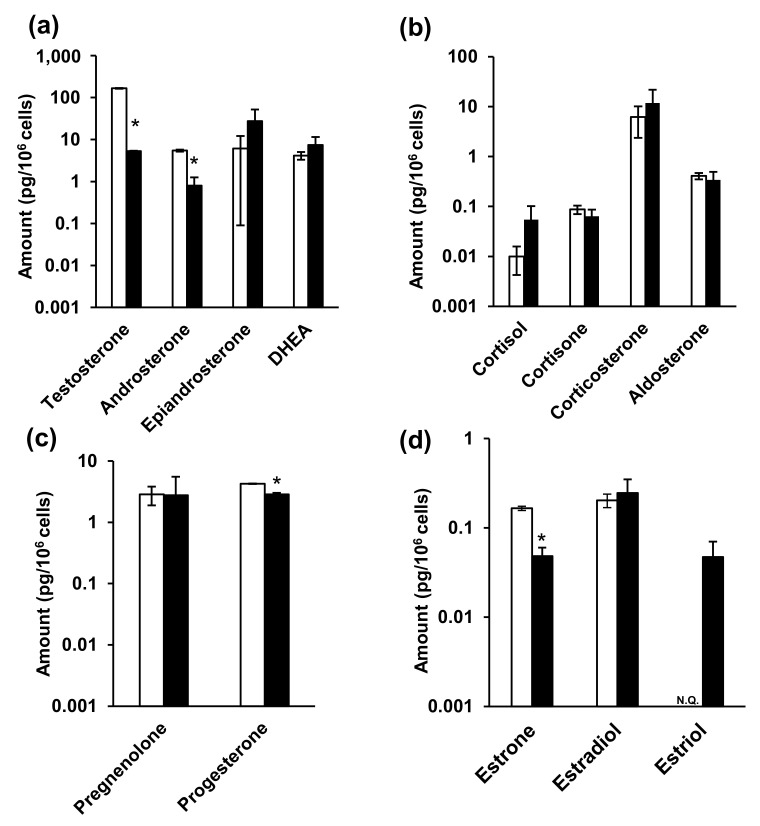
The amounts of steroid hormones in the cells. (**a**) Androgens, (**b**) glucocorticoids and mineralocorticoid, (**c**) progestins, and (**d**) estrogens. White and black bars indicate wild-type and NPC model cells, respectively. * indicates significant differences observed using Wilcoxon’s test (*p* < 0.05). NPC, Niemann–Pick disease type C; N.Q., not quantified.

**Figure 5 ijms-23-04459-f005:**
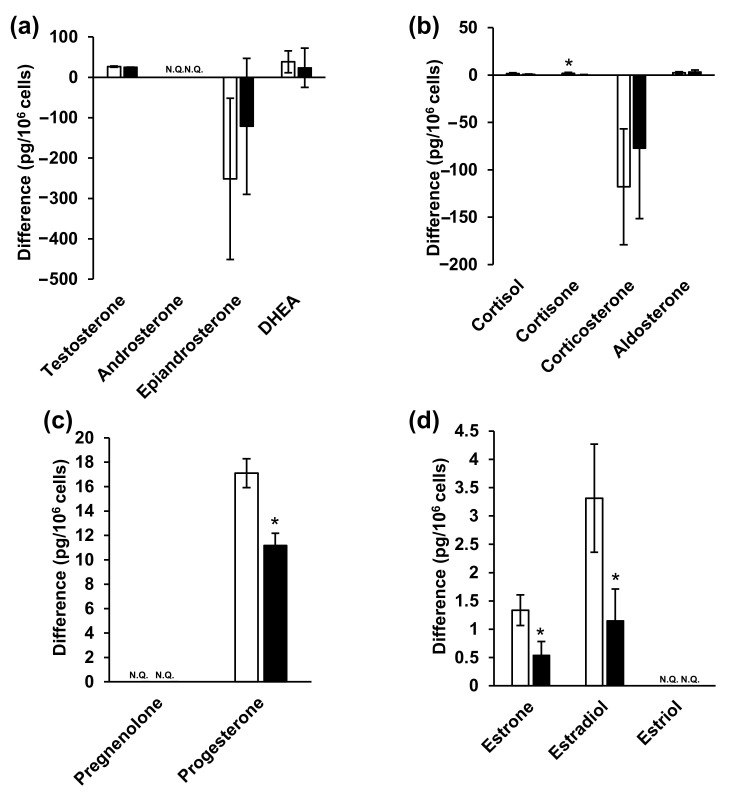
Steroid hormone concentrations in the medium. (**a**) Androgens, (**b**) glucocorticoids and mineralocorticoid, (**c**) progestins, (**d**) estrogens. White and black bars indicate wild-type and NPC model cells, respectively. * indicates significant differences observed using Wilcoxon’s test (*p* < 0.05). NPC, Niemann–Pick disease type C; N.Q., not quantified.

**Figure 6 ijms-23-04459-f006:**
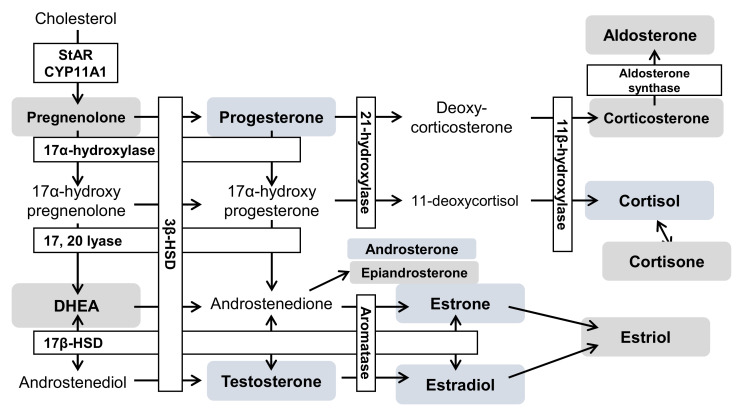
The hypothesis of an altered steroid hormone metabolism in NPC. Blue boxes indicate a decrease; gray boxes indicate no change. The open text indicates the absence of analytes in this study. CYP11A1, cytochrome P450 family 11 subfamily A member 1; 3β-HSD, 3β-hydroxysteroid dehydrogenase; 17β-HSD, 17β-hydroxysteroid dehydrogenase; CYP11B1, 11β-hydroxylase; CYP11B2, aldosterone synthase; CYP17A, 17α-hydroxylase/17,20 lyase; CYP19A, aromatase; CYP21A, 21-hydroxylase; DHEA, dehydroepiandrosterone; StAR, steroidogenic acute regulatory protein.

**Table 1 ijms-23-04459-t001:** SRM parameters and LC retention times for steroid hormones.

Analytes	Q1 (*m*/*z*)	Precursor Ion Form	Q3 (*m*/*z*)	DP (V)	CE (V)	CXP (V)	EP (V)	Polarity	Retention Time (min)
Testosterone	289.0	[M + H]^+^	97.1	100	8	29	6	Positive	7.1
Androsterone	273.1	[M − H_2_O + H]^+^	255.1	146	8	19	14	Positive	10.1
Epiandrosterone	273.2	[M − H_2_O + H]^+^	255.2	101	8	19	14	Positive	8.4
DHEA	271.0	[M − H_2_O + H]^+^	253.1	81	12	19	4	Positive	7.8
Cortisol	363.0	[M + H]^+^	120.9	100	8	31	8	Positive	4.1
Cortisone	361.1	[M + H]^+^	163.0	100	8	33	10	Positive	3.9
Corticosterone	347.2	[M + H]^+^	329.0	101	6	21	16	Positive	5.6
Aldosterone	361.1	[M + H]^+^	343.1	61	8	25	6	Positive	3.3
Pregnenolone	299.2	[M − H_2_O + H]^+^	91.1	161	8	71	8	Positive	10.6
Progesterone	315.0	[M + H]^+^	97.0	100	8	27	6	Positive	10.3
Estrone	269.0	[M − H]^−^	144.8	−80	−6	−48	−9	Negative	6.4
Estradiol	271.0	[M − H]^−^	182.9	−100	−15	−58	−11	Negative	6.1
Estriol	287.1	[M − H]^−^	170.8	−140	−15	−50	−11	Negative	3.0
Testosterone-^2^H_3_	292.1	[M + H]^+^	96.9	100	8	27	6	Positive	7.1
Androsterone-^2^H_4_	277.2	[M − H_2_O + H]^+^	259.1	121	8	19	14	Positive	10.0
DHEA-^2^H_6_	276.9	[M − H_2_O + H]^+^	259.0	211	12	25	11	Positive	7.8
Cortisol-^13^C_3_	366.2	[M + H]^+^	123.9	101	8	31	8	Positive	4.1
Aldosterone-^2^H_7_	368.3	[M + H]^+^	350.0	91	8	25	16	Positive	3.3
Pregnenolone-^13^C_2_,^2^H_2_	303.2	[M − H_2_O + H]^+^	91.1	161	8	71	8	Positive	10.6
Progesterone-^2^H_9_	324.2	[M + H]^+^	100.1	100	8	29	8	Positive	10.2
Estrone-^2^H_4_	273.0	[M − H]^−^	146.9	−80	−6	−52	−11	Negative	6.4
Estradiol-^13^C_3_	274.2	[M − H]^−^	186.0	−100	−15	−56	−9	Negative	6.1
Estriol-^13^C_3_	290.0	[M − H]^−^	173.9	−140	−15	−50	−11	Negative	3.0

## Data Availability

Not applicable.

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
