# Peer review of "Metabolic Alteration Analysis of Steroid Hormones in Niemann–Pick Disease Type C Model Cell Using Liquid Chromatography/Tandem Mass Spectrometry"

_ijms, 2022, doi:10.3390/ijms23084459_

Round 1

Reviewer 1 Report

In my opinion, the authors presented their work with professional high-standard. I have no problem with the construction of the research and the presentation. I only found some minuscule errors:

-line 95: Figure S1, this figure is in the paper and not in the supplementary, so the S is redundant.

-l.194: "stably" is mispronounced.

-l.293-294: after Figure 5. there is no space between the caption and the text.

Author Response

In my opinion, the authors presented their work with professional high-standard. I have no problem with the construction of the research and the presentation. I only found some minuscule errors:
> Thank you four kind review and evaluation. We revised our manuscript.

-line 95: Figure S1, this figure is in the paper and not in the supplementary, so the S is redundant.
> Thank you four kind review and evaluation. We revised the point. Would you please check it (line 84)?

-l.194: "stably" is mispronounced.
> Thank you four kind review and evaluation. We revised the point. Would you please check it (line 160)?

-l.293-294: after Figure 5. there is no space between the caption and the text.
> Thank you four kind review and evaluation. We revised the point. Would you please check it (Between line 236-237)?

Reviewer 2 Report

The paper "Metabolic alteration analysis of steroid hormones in Niemann–Pick disease type C model cell using liquid chromatography/tandem mass spectrometry " by   Ai Abe etal. is too detailed in the methodological aspects. It becomes very heavy to read; on the other hand, the pathophysiological aspects acquire low importance. It is necessary to reduce the text by at least 50%. The language should be reviewed by a native English speaker.

Author Response

The paper "Metabolic alteration analysis of steroid hormones in Niemann–Pick disease type C model cell using liquid chromatography/tandem mass spectrometry " by   Ai Abe etal. is too detailed in the methodological aspects. It becomes very heavy to read; on the other hand, the pathophysiological aspects acquire low importance. It is necessary to reduce the text by at least 50%.

> Thank you for your kind review and evaluation. As your point out, we revised the manuscript and reduced the text drastically. We also attached highlighted manuscript for understanding revised points. Would you please check it?

The language should be reviewed by a native English speaker.

> Thank you for your kind review. The paper was reviewed and revised with a native English speaker. Would you please check it?

Round 2

Reviewer 2 Report

Accept as it is